# The *HINT1* Gene rs2526303 Polymorphism and Its Association with Personality Traits in Cigarette Smokers

**DOI:** 10.3390/ijms25021218

**Published:** 2024-01-19

**Authors:** Aleksandra Suchanecka, Agnieszka Boroń, Krzysztof Chmielowiec, Aleksandra Strońska-Pluta, Jolanta Masiak, Milena Lachowicz, Jolanta Chmielowiec, Joanna Janiszewska-Olszowska, Anna Grzywacz

**Affiliations:** 1Independent Laboratory of Health Promotion, Pomeranian Medical University in Szczecin, Powstańców Wielkopolskich 72 St., 70-111 Szczecin, Poland; aleksandra.suchanecka@pum.edu.pl (A.S.); aleksandra.stronska@pum.edu.pl (A.S.-P.); 2Department of Clinical and Molecular Biochemistry, Pomeranian Medical University in Szczecin, Powstańców Wielkopolskich 72 St., 70-111 Szczecin, Poland; agnieszka.boron@pum.edu.pl; 3Department of Hygiene and Epidemiology, Collegium Medicum, University of Zielona Góra, 28 Zyty St., 65-046 Zielona Góra, Poland; k.chmielowiec@inz.uz.zgora.pl (K.C.); j.chmielowiec@inz.uz.zgora.pl (J.C.); 4Second Department of Psychiatry and Psychiatric Rehabilitation, Medical University of Lublin, 1 Głuska St., 20-059 Lublin, Poland; jolanta.masiak@umlub.pl; 5Department of Psychology, Gdansk University of Physical Education and Sport, 80-336 Gdansk, Poland; milena.lachowicz@awf.gda.pl; 6Department of Interdisciplinary Dentistry, Pomeranian Medical University in Szczecin, 70-111 Szczecin, Poland; joanna.janiszewska.olszowska@pum.edu.pl

**Keywords:** nicotine addiction, *HINT1* gene, personality, anxiety

## Abstract

The development of a substance use disorder (SUD) is a multifaceted process influenced by both genetic and environmental factors. Recent research has suggested the potential involvement of the *HINT1* gene in various aspects of plasticity, mood regulation, anxiety-like behaviour, and stress-coping mechanisms. Moreover, personality traits are also recognised to be instrumental in developing substance dependency. Given these considerations, our study investigated the associations among cigarette smoking, personality traits, and the rs2526303 polymorphism. Additionally, we investigated the interactions between personality traits and rs2526303 in the *HINT1* gene. The study group comprised 531 volunteers: 375 cigarette users (mean age = 29.42 ± 10.72; F = 49%, M = 51%) and 156 never-smokers (mean age = 26.93 ± 10.09; F = 79%, M = 21%). Genotyping was conducted using the real-time PCR method, and the NEO Five-Factor Personality Inventory and State–Trait Anxiety Inventory were administered. There were no statistically significant differences in the frequency of rs2526303 genotypes and alleles in the cigarette user group compared to the control group. Compared to the control group, the cigarette users obtained higher scores in the assessment of the NEO-FFI Extraversion scale and lower results for the NEO-FFI Openness, Agreeableness, and Conscientiousness scales. Additionally, there was a statistically significant effect of rs2526303 genotype interaction and cigarette-using status on the conscientiousness scale. These outcomes collectively suggest a notable association between cigarette smoking and specific dimensions of personality, particularly highlighting differences in extraversion, openness, agreeableness, and conscientiousness. Furthermore, the detected interaction effect involving rs2526303 concerning conscientiousness signifies a complex interplay between genetic factors and smoking behaviour.

## 1. Introduction

Substance use disorder (SUD) is a multifaceted condition resulting from genetic and environmental factors. Recent research has identified potential genes involved in the risk of developing SUDs through genome-wide association studies [1,2,3,4,5]. SUDs encompass various types of drug use, including smoking, alcohol, prescription medications, and illicit drugs, and they are a significant global health issue. Rates of SUDs vary by substance and country, but the overall prevalence is high [6,7]. Each year, smoking, alcohol, and illicit drug use collectively cause the death of 11.8 million people. Men are more likely to have SUDs, and over half of alcohol and drug overdose deaths occur in people under 50. Approximately 35 million individuals worldwide require treatment for SUDs [8].

The human histidine triad nucleotide-binding protein 1 (hHint1) belongs to the histidine triad (HIT) protein superfamily. It is encoded by the *HINT1* gene located on chromosome 5q31.22 in a region linked to schizophrenia in genetic studies [9,10,11]. *HINT1* is expressed ubiquitously and forms homodimeric purine phosphoramides comprising 126 amino acids. It is one of three HINT proteins (HINT1, HINT2, and HINT3) in the human genome. HINT1 regulates transcriptional and cell cycles [12]. Although *HINT1* is expressed widely in the liver, kidney, and brain, including the mesocortical and mesostriatal regions [13], its physiological function must be better understood. However, microarray analysis has identified the gene as a potential contributor to the neuropathology of schizophrenia [14,15]. Expression studies have found that *HINT1* mRNA levels are significantly reduced in the prefrontal cortex (PFC) of males with schizophrenia compared to controls [16]. Notably, *HINT1* also appears to modulate the effects of drug abuse. Research on the CNS function of HINT1 has revealed that the protein interacts specifically with the C-terminus of the μ-opioid receptor. The absence of HINT1 leads to decreased receptor desensitization and the inhibition of PKC-mediated μ-opioid receptor phosphorylation [17]. This can result in enhanced basal and morphine-induced antinociception, improved morphine tolerance, and increased sensitivity to the effects of amphetamine and apomorphine. Studies by Liu et al. (2021) [18] have shown that the HINT1 protein is involved in various stages of addiction and may attenuate morphine addiction behaviour and withdrawal symptoms. In addition, studies using *HINT1* knockout mice suggest that HINT1 may play a role in antidepressant and anxiety-like behaviours [19,20]. A study by Jackson et al. (2011) [21] found that genetic variation in the *HINT1* gene is protective against nicotine dependence (ND), and the levels of the HINT1 protein in the nucleus accumbens are changed after prolonged exposure to nicotine, suggesting a role for *HINT1* in nicotine-mediated responses. *Post-mortem* mRNA expression in humans reveals that smoking status and genotype affect *HINT1* expression in the brain. Animal studies demonstrated increased HINT1 protein levels in the mouse nucleus accumbens (NAc) after chronic nicotine exposure [21]. Data from the same team [22] regarding *HINT1* knockout mice further support a role for *HINT1* in nicotine-mediated behaviours. It is suggested that alterations in the gene may have gender-specific effects on the phenotype.

According to research conducted by Castelli et al. in 2023 [23], exposure to tetrahydrocannabinol (THC) during prenatal development can lead to specific difficulties with spatial cognitive processing and impair the effectors of hippocampal neuroplasticity. It was found that HINT1 plays a role in this process. Research has shown that social isolation in mice can cause behaviours associated with schizophrenia, including social withdrawal, anxiety disorders, cognitive issues, and sensorimotor gating problems. These changes are linked to alterations in genes such as *HINT1*. Social isolation is a reliable model for studying how early-life stress can affect schizophrenia-related behaviour in mice [24]. Additionally, early-life stress may contribute to the development of addiction as a maladaptive coping mechanism. HINT1 was identified for the stratum pyramidale in females in a study analysing hippocampal neuroanatomy correlating with spatial learning ability in mice [25], pointing to its role in neuroplasticity, another factor influencing and influenced by substance abuse. The studies mentioned above support a possible role for the *HINT1* gene in plasticity, mood regulation, anxiety-like behaviour, and stress-coping mechanisms.

Personality is another factor that influences the development and course of substance dependency, in addition to genetics. The Five-Factor Model of Personality (FFM) is widely recognised as the standard for classifying personality traits [26,27]. The FFM was developed by analysing trait adjectives in English using questionnaires and statistical methods such as factor analysis [26,28]. The FFM comprises five broad traits, which are commonly called the Big Five: neuroticism, agreeableness, extraversion, openness, and conscientiousness [29]. The FFM has been translated into several languages, and research shows that the traits reflect fundamental individual differences across various cultures [30,31]. This suggests that these five traits may have biological and evolutionary foundations [32,33]. The NEO Personality Inventory—Revised (NEO-PI-R) is a widely used tool for assessing the Five-Factor Model (FFM) of personality traits, as developed by Costa and McCrae in 1995 [34]. The NEO-PI-R breaks down the five global traits into six underlying facets. This instrument has been translated into multiple languages and studied across diverse populations, such as with a validation study conducted on an American Substance Use Disorder (SUD) cohort by Piedmont and Ciarrocchi in 1999 [35]. These studies have found the NEO-PI-R to have good internal consistency, stable factor loadings, high test–retest reliability, and good content and criterion validity. The FFM traits are known to be correlated and can be sorted into two meta-traits: stability (beta), which reflects the shared variance of neuroticism (inversely associated), agreeableness, and conscientiousness, and plasticity (alpha), which encompasses the shared variance of openness and extraversion [36]. Additionally, a general factor of personality—known as the Big One—has been suggested for scores for openness, conscientiousness, agreeableness, and extraversion, which are inversely and positively associated with neuroticism [37]. Individuals with substance use disorders often exhibit higher levels of neuroticism and lower levels of both conscientiousness and agreeableness compared to those without SUDs, as indicated by various assessments of the Five-Factor Model of personality [38,39,40].

Our study aimed to analyse the relationships between cigarette smoking, the personality traits measured with the NEO-FFI and anxiety measured with the STAI, and rs2526303. Additionally, we investigated the association among personality traits, anxiety, and polymorphism rs2526303 in the *HINT1* gene.

## 2. Results

The frequency of allele and genotype distributions agreed with the HWE in the cigarette users and control subjects (Table 1).

There were no statistically significant differences in the frequency of rs2526303 genotypes in the tested cigarette users compared to the control group (T/T 0.14 vs. T/T 0.17; C/T 0.46 vs. C/T 0.51; C/C 0.40 vs. C/C 0.33, χ^2^ = 2.4540, *p* = 0.2931). There were no significant differences in the frequency of rs2526303 alleles between cigarette users and the control group (T 0.37 vs. T 0.42; C 0.63 vs. C 0.57, χ^2^ = 2.2528, *p* = 0.1334) (Table 2).

The means and standard deviations for the NEO-FFI and the STAI scales for the cigarette users and controls are shown in Table 3. Compared to the control group, the cigarette users obtained higher scores in the assessment of the NEO-FFI Extraversion scale (5.96 vs. 5.33; Z = 2.9893; *p* = 0.0027) and lower results for the NEO-FFI Openness scale (5.21 vs. 5.61; Z = −2.4160; *p* = 0.0157), the NEO-FFI Agreeableness scale (5.27 vs. 6.26; Z = −4.3719; *p* ≤ 0.0000), the NEO-FFI Conscientiousness scale (5.83 vs. 6.65; Z = −3.7791; *p* = 0.0002). 

The results of Friedman’s ANOVA of the NEO Five-Factor Personality Inventory and the State–Trait Anxiety Inventory sten scores are summarised in Table 4. There was a statistically significant effect of the rs2526303 genotype interaction and cigarette-using status on the Conscientiousness scale (T1 = 14.36; *p* = 0.01345; when accepting missing values, Durbin/Skillings–Mack = 25.52; *p* = 0.00011; Figure 1). 

In the pot hoc analysis, cigarette users with the TT genotype had a significantly higher score on the Conscientiousness scale than that of cigarette users with the CC genotype (*p* = 0.03323, Table 5). Cigarette users with the CT genotype had a significantly lower level on the Conscientiousness scale than that of controls with the CT genotype (*p* = 0.01328, Table 5). Cigarette users with the CC genotype had a significantly lower level on the Conscientiousness scale compared to that of controls with the CT (*p* = 0.00055, Table 5) and CC genotypes (*p* = 0.0107, Table 5). The control group with the TT genotype had a significantly lower level on the Conscientiousness scale than that of the control group with the CT genotype (*p* = 0.03657, Table 5). 

## 3. Discussion

Our study investigated the associations between cigarette smoking, personality traits, anxiety, and the rs2526303 polymorphism. Additionally, we investigated the interactions between the variables mentioned above.

We found no statistically significant differences in the frequency of rs2526303 genotypes and alleles in the tested cigarette users compared to the controls. However, a statistically significant effect of the rs2526303 genotype interaction and cigarette-using status on the NEO-FFI Conscientiousness scale was observed. The analysed single-nucleotide polymorphism rs2526303 is located in the non-coding region of the *HINT1* gene and has been investigated before in two studies regarding schizophrenia [41] and nicotine dependence [21]. The analysed SNP was significant in the male schizophrenia sample. There has been a recent discussion around the potential for sex to have a more powerful influence on brain structure and its relationship to psychiatric disorders than is commonly acknowledged [42,43,44]. In our sample, we did not discover significant differences in frequencies of alleles and genotypes in a sex-stratified analysis. 

In the central nervous system, *HINT1* immunoreactivity is located in neurons and neuronal processes. Martins-de-Souza et al. (2012) [45] found that the levels of HINT1 were increased in major depressive disorder (MDD) patients. Interestingly, the protein levels were also decreased in previous studies of schizophrenia [20]. HINT1 was described as a PKC inhibitor, but its precise function is not fully understood. It is widely expressed in several tissues, including the liver, kidneys, and brain, where it has been implicated in controlling transcriptional processes, tumour suppression, and susceptibility [46]. A potential role for HINT1 in MDD and anxiety disorders was recently described. This was based on studies using *HINT1* knockout (KO) mice. Lack of *HINT1* expression altered striatal and nucleus accumbens postsynaptic dopamine transmission and increased circulating corticosterone levels, suggesting effects on the hypothalamic–pituitary–adrenal axis. In addition, this model showed anxiety-like behaviour [20,47]. These findings indicate that HINT1 may be a novel biomarker for MDD and may be specific for subjects without psychosis. This was demonstrated by the fact that this protein was only found to be elevated in people with MDD who did not develop psychosis. These findings suggest that the *HINT1* gene may be crucial in regulating mood in the central nervous system. The histidine triad nucleotide-binding protein-1 gene is implicated in schizophrenia and the behavioural effects of morphine [17] and amphetamine [16]. Because nicotine dependence is highly comorbid with schizophrenia and other substance abuse, Jackson et al. (2011) [21] examined the association of *HINT1* with nicotine dependence. The study showed that two markers (rs3864283 and rs2526303) were significantly associated with ND—specifically, with the Fagerström Test of Nicotine Dependence (FTND) scores and the number of cigarettes smoked per day (numCIG), suggesting that the gene was associated with an increased risk of ND. In addition, *HINT1* mRNA expression levels in the brain tended toward higher expression in smokers than in non-smokers. However, these results did not reach significance due to the small sample size. Interestingly, the main effects of the genotype suggested that individuals with the specific polymorphism, which was found to be associated with the FTND score and numCIGs, had significantly higher *HINT1* expression levels. In contrast, the genotype (rs3864283) × smoking status interaction suggested that this effect was dependent on the smoking status; that is, smokers with the risk allele of rs3864283 had higher *HINT1* expression than that in non-smokers. Additionally, subjects with the major alleles of rs3864283 and rs2526303 (T and C, respectively, for the two markers) had higher FTND scores and numCIG than those with the minor alleles did. The authors concluded that variants of the *HINT1* gene are associated with pathomechanisms of nicotine dependence, as a change in mRNA expression in smokers may suggest that the polymorphism has some biological relevance in the development of ND. Studies conducted on mice demonstrated increased HINT1 protein levels in the nucleus accumbens (NAc) after chronic nicotine exposure. This increase was reduced after treatment with the nicotinic receptor antagonist [21]. In a study by Jackson et al. (2012) [22], using a set of behavioural tests, the role of *HINT1* in acute behaviour mediated by nicotine was elucidated with male and female *HINT1* wild-type and knockout (KO) mice. The results showed that male *HINT1* KO mice exhibited reduced sensitivity to acute nicotine-induced antinociception in the tail-flick test but not in the hot plate test. At low doses of nicotine, both male and female *HINT1* KO mice showed reduced sensitivity to nicotine-induced hypomotility, but the effect was more significant in females. The initial differences in locomotor activity seen in male *HINT1* wild-type and KO mice were not observed in females. Nicotine did not induce an anxiolytic effect in male *HINT1* KO mice but instead induced an anxiogenic response. Diazepam was also unable to induce an anxiolytic effect in these mice, suggesting a general anxiety phenotype that was not dependent on nicotine. No significant differences in anxiety-like behaviour were detected in female mice. 

We found statistically significant differences between smokers and controls regarding personality traits measured with the NEO-FFI, i.e., the score for the Extraversion trait was higher in smokers, and the Openness, Agreeableness, and Conscientiousness scores were lower in smokers than in the controls. Subjects with substance use disorders tend to score higher on the Neuroticism scale and lower on the Conscientiousness and Agreeableness scales compared to individuals without SUDs across different measures of the Five-Factor Model [38,39,40,48]. This SUD profile is equivalent to a low score on the meta-trait stability [36]. Kotov et al. (2010) [40] found that several psychiatric disorders were linked to high Neuroticism. Interestingly, in our study, this trait was not significant. Conscientiousness, a trait that describes a person’s tendency to plan and organise [26], is also linked to several psychiatric disorders [40]. Mental distress may deplete the psychological capacity to plan and organise, or individuals with low Conscientiousness may be more vulnerable to the development of psychiatric disorders after stressful life experiences. Moreover, a phenomenological overlap exists between Conscientiousness and executive functioning, and it is diminished in individuals with SUDs [49]. Agreeableness reflects interpersonal inclinations to be altruistic and the assumption that others will be too [28]. Low Agreeableness is specific only to SUDs and no other psychiatric conditions [40]. Reduced motivation and capacity to sustain positive relationships with others could result from high levels of childhood trauma [50] and the comorbidity of ADHD [51] found in patients with SUDs. A Norwegian opioid-dependence population showed low scores for Extraversion [52]. Nevertheless, studies of other substance-dependent and substance-using populations have found higher scores on the Extraversion scale [38,39,53] and its facet of excitement-seeking [54,55]. The variability in the results for Extraversion in people with SUDs may, in part, be due to different personality inventories used in the studies or may result from an underlying difference in personality between different SUD subgroups (type of substance used, frequency and severity of use, and gender). There is less variability regarding findings on the Openness trait. Still, higher scores on the Openness scale have been reported in student users of cannabis [53,56], and lower levels of Openness have been reported among students with problematic alcohol patterns [57]. 

We found no statistically significant differences regarding anxiety measures for both state and trait facets analysed with the State–Trait Anxiety Inventory in cigarette smokers when compared to the controls. Nevertheless, studies have indicated a positive association between smoking and mental illness, with smoking rates rising with the severity of the disease [58]. People who experience mental illness tend to initiate smoking at an earlier age, smoke more cigarettes, and become more heavily dependent on cigarettes than the general population. For instance, a survey performed in England indicated that individuals with mental illness are responsible for 42% of all cigarette consumption. However, this also includes substance use disorders [59]. Furthermore, whilst the consumption of cigarettes in the overall population has experienced a steady decline over the last two decades, consumption amongst individuals with mental illness has remained relatively stable [60]. Most of the studies examining the link between baseline anxiety and smoking onset found evidence of an association with an increased risk of initiation [61,62,63,64], and one study reported opposite results [65]. A study by Kandel et al. (2007) [66] analysing a bidirectional relationship between smoking anxiety in sex-stratified groups found that it was associated with later smoking behaviour only in females. A systematic review performed by Fluharty et al. (2017) [67] showed that baseline anxiety or depression was associated with some later smoking behaviour (onset of smoking itself, increased smoking heaviness, or the transition from daily smoking into dependence) in nearly half of the analysed studies, which supports the self-medication model of nicotine usage [68,69]. The alternative model of nicotine use and mental illness states that prolonged smoking elevates susceptibility to depression and anxiety [70,71]. Additionally, the associations observed between smoking and mental health may be a result of shared genetic and environmental factors.

Our study focused solely on individuals of Caucasian descent, necessitating validation of our findings across diverse populations. Furthermore, our analysis concentrated on a single SNP within the *HINT1* gene, limiting the scope of our conclusions based on this specific dataset. To augment our understanding, future investigations will encompass methylation analysis across a larger cohort containing various substance addictions, enabling a more comprehensive exploration of the *HINT1* gene’s role.

## 4. Materials and Methods

### 4.1. Participants

The study group comprised 531 volunteers. Of these, 375 were cigarette users (mean age = 29.42, SD = 10.72; F = 49%, M = 51%) and 156 were never-smokers (mean age = 26.93, SD = 10.09; F = 79%, M = 21%). The research was conducted at the Independent Laboratory of Health Promotion, Pomeranian Medical University in Szczecin. The study group volunteered on the basis of an announcement and social media posts. The subjects were mainly from the West Pomeranian Voivodeship, Poland. All participants were validated for eligibility based on their smoking status, and subjects aged 18–60 years were included. The study did not include subjects who had neuropsychiatric disorders or dependencies on substances other than nicotine, who were pregnant or breastfeeding, who were under 18 or over 60 years of age, or who had intellectual disabilities, metabolic diseases, or serious medical conditions. Both groups were examined with the NEO Five-Factor Personality Inventory (NEO-FFI) and State–Trait Anxiety Inventory (STAI). All participants gave written informed consent prior to entering the study. Approval was obtained from the Bioethical Committee of the Pomeranian Medical University in Szczecin (KB-0012/164/17-A).

### 4.2. Psychometric Tests

The State–Trait Anxiety Inventory assesses anxiety, which can be defined as a continuous tendency to experience worry, stress, and discomfort, as well as anxiety states such as fear and temporary activation of the autonomic nervous system prompted by particular situations, as a trait [72].

The Five-Factor Inventory comprises six components for each of the following five traits: neuroticism (susceptibility to stress, depression, anxiety, self-awareness, impulsivity, hostility), extroversion (warmth, assertiveness, positive emotions, activity, sociability, emotion seeking), openness to experience (values, aesthetics, feelings, fantasy, actions, ideas), agreeableness (straightforwardness, modesty, altruism, compliance, tenderness, trust), and conscientiousness (order, self-discipline, competence, striving for achievements, consideration, duty) [73]. 

The NEO-FFI and STAI findings were presented as sten scores. The transformation of the raw score to the sten scale was conducted according to the Polish standards [74,75] for adults, where it was assumed that 1–2 corresponded to very low scores, 3–4 to low scores, 5–6 to average scores, 7–8 to high scores, and 9–10 to very high scores. 

### 4.3. Genotyping

The genomic DNA was extracted from venous blood using a commercially available protocol (QIAamp Blood DNA Mini Kit, QIAGEN, Hilden, Germany). Genotyping was performed by using the real-time PCR method. The fluorescence signal was plotted with a temperature function, and melting curves for each sample were obtained. The *HINT1* gene rs2526303 peaks were read at 62.31 °C for the T allele and 69.26 °C for the C allele.

### 4.4. Statistical Analysis

The concordance between the genotype distribution and Hardy–Weinberg Equilibrium (HWE) was tested using the HWE software (https://wpcalc.com/en/equilibrium-hardy-weinberg/ (accessed on 5 April 2023). The relations between rs2526303 variants, cigarette users and control subjects, and the NEO Five-Factor Inventory were analysed using Friedman’s ANOVA, and a test of accepting missing values was also used (Durbin/Skillings–Mack). The condition of homogeneity of variance was fulfilled (Levene test *p* > 0.05). The analysed variables were not normally distributed. The NEO Five-Factor Inventory scores (Neuroticism, Extraversion, Openness, Agreeability, and Conscientiousness) were compared in the studied groups using the Mann–Whitney U-test. The differences in rs2526303 genotype and allele frequencies in control subjects and cigarette users were tested using the chi-square test. All computations were performed using STATISTICA 13 (Tibco Software Inc., Palo Alto, CA, USA) and PQStat 1.8.6.102 (PQStat Software, Poznan, Poland) for Windows 10 (Microsoft Corporation, Redmond, WA, USA).

## 5. Conclusions

Our study investigated the relationships between cigarette smoking, personality traits, anxiety, and the rs2526303 polymorphism within the *HINT1* gene. Additionally, we investigated the associations between personality traits, anxiety, and the rs2526303 polymorphism in the *HINT1* gene. Despite no statistically significant differences being observed in the frequency of rs2526303 genotypes and alleles between the cigarette users and the control group, distinct associations emerged between cigarette use and certain personality traits. The findings indicated that individuals who were cigarette users exhibited higher scores in the assessment of the NEO-FFI Extraversion scale compared to the control group. Conversely, lower results were observed in the Openness, Agreeableness, and Conscientiousness scales of the NEO-FFI among cigarette users compared to non-smokers in the control group. Moreover, an intriguing aspect of our investigation revealed a statistically significant effect of the rs2526303 genotype interaction in conjunction with cigarette-using status on the Conscientiousness scale. This interaction implies a potential nuanced influence of this genetic variation in shaping the relationship between cigarette use and conscientious personality traits. Future perspectives for investigating the *HINT1* gene in nicotine-using individuals include the analysis of epigenetic changes, i.e., the methylation of *HINT1* promoter CpG islands and gene expression studies. 

These outcomes collectively suggest a notable association between cigarette smoking and specific dimensions of personality, particularly highlighting differences in extraversion, openness, agreeableness, and conscientiousness. Furthermore, the detected interaction effect involving rs2526303 concerning conscientiousness signifies a complex interplay between genetic factors and smoking behaviour.

## Figures and Tables

**Figure 1 ijms-25-01218-f001:**
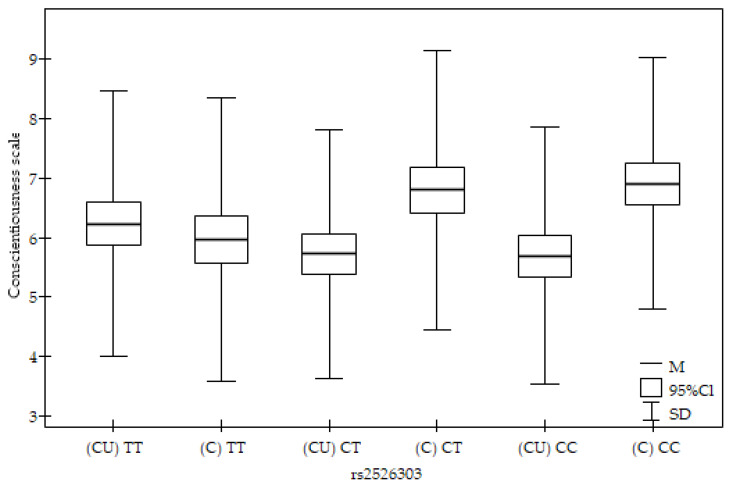
Associations between smoking status in cigarette users (CU)/control (C), *HINT1* rs2526303, and the NEO-FFI Conscientiousness scale.

**Table 1 ijms-25-01218-t001:** Hardy–Weinberg’s equilibrium for *HINT1* rs2526303 in a group of cigarette users and control subjects.

Hardy–Weinberg Equilibrium, Including Analysis for Ascertainment Bias		Observed (Expected)	Allele Freq	χ^2^ (*p*-Value)
rs2526303 cigarette users n = 375	T/T	52 (51.5)	p (C) = 0.63 q (T) = 0.37	0.0112 (0.9158)
	C/T	174 (175.0)		
	C/C	149 (148.5)		
rs2526303 control n = 156	T/T	26 (16.2)	p (C) = 0.58 q (T) = 0.42	0.2436 (0.6216)
	C/T	79 (68.2)		
	C/C	51 (66.5)		

*p*—statistical significance according to the χ^2^ test.

**Table 2 ijms-25-01218-t002:** The *HINT1* rs2526303 genotypes and alleles and their frequencies in the cigarette users and control subjects.

	rs2526303				
	Genotypes			Alleles	
	T/T	C/T	C/C	T	C
	n (%)	n (%)	n (%)	n (%)	n (%)
Cigarette Users n = 375	52	174	149	278	472
	(13.87%)	(46.40%)	(39.73%)	(37.07%)	(62.93%)
Control	26	79	51	131	181
n = 156	(16.67%)	(50.64%)	(32.69%)	(41.99%)	(57.01%)
χ^2^	2.4540			2.2528	
(*p*-value)	(0.2931)			(0.1334)	

*n*—number of subjects.

**Table 3 ijms-25-01218-t003:** STAI and NEO Five-Factor Inventory sten scores for cigarette users and controls.

STAI/NEO Five-Factor Inventory	Cigarette Users (n = 375)	Control (n = 156)	Z	(*p*-Value)
STAI trait scale	5.95 ± 2.45	5.70 ± 2.14	1.5845	0.1131
STAI state scale	5.55 ± 2.35	5.48 ± 2.20	0.6304	0.5284
Neuroticism scale	5.92 ± 2.23	5.71 ± 1.95	1.4710	0.1399
Extraversion scale	5.96 ± 2.08	5.33 ± 1.94	2.9893	0.0027 *
Openness scale	5.21 ± 2.02	5.61 ± 2.04	−2.4160	0.0157 *
Agreeability scale	5.27 ± 2.26	6.26 ± 2.38	−4.3719	0.0000 *
Conscientiousness scale	5.83 ± 2.11	6.65 ± 2.28	−3.7791	0.0002 *

*p*, statistical significance with the Mann–Whitney U-test; *n*, number of subjects; M ± SD, mean ± standard deviation; * statistically significant difference.

**Table 4 ijms-25-01218-t004:** The results of Friedman’s ANOVA for cigarette users and controls for the NEO Five-Factor Inventory, STAI, and *HINT1* rs2526303.

STAI/NEO Five-Factor Inventory	Group	rs2526303			ANOVA Friedman	
		T/T n = 78 M ± SD	C/T n = 271 M ± SD	C/C n = 216 M ± SD	T1 Friedman Statistic T_1_ (*p*-Value) df = 5	Accepting Missing Values; Durbin/Skillings–Mack (*p*-Value) df5
STAI state scale	Cigarette Users (CU); n = 375	5.75 ± 2.43	5.31 ± 2.43	5.78 ± 2.22	T_1_ = 7.35 (*p* = 0.19548)	8.02 (*p* = 0.15507)
	Control; n = 156	5.15 ± 2.22	5.38 ± 1.94	5.84 ± 2.58		
STAI trait scale	Cigarette Users (CU); n = 375	5.86 ± 2.57	5.72 ± 2.48	6.27 ± 2.29	T_1_ = 3.48 (*p* = 0.62687)	6.35 (*p* = 0.27385)
	Control; n = 156	5.42 ± 1.32	5.43 ± 2.05	6.08 ± 2.50		
Neuroticism scale	Cigarette Users (CU); n = 375	5.80 ± 2.47	5.90 ± 2.22	6.00 ± 2.12	T_1_ = 7.35 (*p* = 0.19548)	8.02 (*p* = 0.15507)
	Control; n = 156	5.65 ± 1.59	5.67 ± 2.10	5.82 ± 1.94		
Extraversion scale	Cigarette Users (CU); n = 375	6.07 ± 1.87	5.84 ± 2.05	6.06 ± 2.21	T_1_ = 3.48 (*p* = 0.62687)	6.35 (*p* = 0.27385)
	Control; n = 156	5.46 ± 1.86	5.57 ± 1.88	4.84 ± 2.03		
Openness scale	Cigarette Users (CU); n = 375	5.00 ± 1.95	5.49 ± 2.12	4.93 ± 1.89	T_1_ = 6.45 (*p* = 0.26519)	9.36 (*p* = 0.09549)
	Control; n = 156	5.58 ± 2.19	5.71 ± 2.04	5.59 ± 1.98		
Agreeability scale	Cigarette Users (CU); n = 375	5.48 ± 2.58	5.24 ± 2.26	5.18 ± 2.14	T_1_ = 7.03 (*p* = 21,813)	18.63 (*p* = 0.00226) *
	Control; n = 156	5.84 ± 2.74	6.44 ± 2.46	6.25 ± 2.06		
Conscientiousness scale	Cigarette Users (CU); n = 375	6.36 ± 2.07	5.77 ± 2.09	5.70 ± 2.15	T_1_ = 14.36 (*p* = 0.01345) *	25.52 (*p* = 0.00011) *
	Control; n = 156	5.96 ± 2.37	6.79 ± 2.34	6.90 ± 2.11		

*—significant result; CU—cigarette users; M ± SD—mean ± standard deviation.

**Table 5 ijms-25-01218-t005:** Conover–Iman post hoc analysis of interactions between the smoking status (cigarette users/control), *HINT1* rs2526303, and the NEO-FFI Conscientiousness scale.

rs2526303 and Conscientiousness Scale						
	{1} M = 6.37	{2} M = 5.77	{3} M = 5.70	{4} M = 5.96	{5} M = 6.80	{6} M = 6.90
Cigarette Users T/T {1}		0.26639	0.03323 *	0.47429	0.16534	0.66172
Cigarette Users C/T {2}			0.30190	0.69079	0.01328 *	0.12248
Cigarette Users C/C {3}				0.15368	0.00055 *	0.0107 *
Control T/T {4}					0.03657 *	0.24978
Control C/T {5}						0.34046
Control C/C {6}						

*—significant statistical differences, M—mean.

## Data Availability

The genotyping and psychometric tests’ results are available upon request.

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
