# Peer review of "The HINT1 Gene rs2526303 Polymorphism and Its Association with Personality Traits in Cigarette Smokers"

_ijms, 2024, doi:10.3390/ijms25021218_

Round 1
Reviewer 1 Report
Comments and Suggestions for Authors
This article explores the relationships between The HINT1 Gene rs2526303 Polymorphism and the personality characteristics of both smokers and non-smokers (anxiety-trait is also a personality dimension). While the topic is interesting and potentially valuable from a scientific perspective on smoker behavior, there are some methodological issues that, in my view, need clarification and correction.
The objective of the study yielded negative results, as no association was found between the polymorphism and the tobacco-related behaviors of the participants. Subsequently, the authors present a series of correlations between different variables that appear to be cherry-picking results, and it is not clear what the purpose and/or objective of these correlations are. This methodological approach needs clarification and significantly deviates from the scope of this journal, so it would be worth considering publishing this article in a journal more dedicated to psychology and/or psychological characteristics related to addiction.
Specifically, the article raises the following methodological concerns for me:
- The number of variables included and the number of correlations attempted suggest a correction of the p-value for multiple comparisons, which was not performed. It is quite possible that significance may not survive such a mandatory correction. I would like to see these correction performed.
- The use of multivariate analysis of factor effects ANOVA in a sample that does not have a normal distribution is highly questionable from a methodological standpoint. I recommend that the authors, instead, conduct Nonparametric Tests such as the Friedman test.
Addressing these two methodological assumptions is crucial for the results to be analyzed and discussed in a more comprehensive manner.
Comments on the Quality of English Language-
Author Response
Dear Reviewer,
We would like to thank you for your valuable comments on the article. Below you will find our reply to your review. All changes are with a description or a comment, and changes have been made to the manuscript (track changes in the tracking group on the review tab).
Comments and Suggestions for Authors
This article explores the relationships between The HINT1 Gene rs2526303 Polymorphism and the personality characteristics of both smokers and non-smokers (anxiety-trait is also a personality dimension). While the topic is interesting and potentially valuable from a scientific perspective on smoker behavior, there are some methodological issues that, in my view, need clarification and correction.
The objective of the study yielded negative results, as no association was found between the polymorphism and the tobacco-related behaviors of the participants. Subsequently, the authors present a series of correlations between different variables that appear to be cherry-picking results, and it is not clear what the purpose and/or objective of these correlations are. This methodological approach needs clarification and significantly deviates from the scope of this journal, so it would be worth considering publishing this article in a journal more dedicated to psychology and/or psychological characteristics related to addiction.
Specifically, the article raises the following methodological concerns for me:
The number of variables included and the number of correlations attempted suggest a correction of the p-value for multiple comparisons, which was not performed. It is quite possible that significance may not survive such a mandatory correction. I would like to see these correction performed.
The use of multivariate analysis of factor effects ANOVA in a sample that does not have a normal distribution is highly questionable from a methodological standpoint. I recommend that the authors, instead, conduct Nonparametric Tests such as the Friedman test.
Addressing these two methodological assumptions is crucial for the results to be analyzed and discussed in a more comprehensive manner.
Thank you for these valuable comments. We have used suggested statistical tests resulting in changes in Table 4 and Table 5, proper descriptions in the Results and the Methods sections were added.
Reviewer 2 Report
Comments and Suggestions for Authors
Dear editors,
Firstly, I would like to commend the authors on their intriguing and generally well-structured manuscript exploring the correlation between cigarette smoking and personality traits, anxiety, and the HINT1/rs2526303 polymorphism.
However, for this work to meet the standards required for publication, several improvements are essential. I have outlined the key areas that require attention below:
-
Introduction:
-
Addition of citations in specific sections is necessary to strengthen the scholarly foundation of the manuscript.
-
Discussion:
-
An extensive rearrangement of the material is recommended. The authors' findings and their correlation with relevant literature should form the central focus of this section. Extraneous theoretical or irrelevant information should be eliminated.
-
Establishing a clear connection between the conclusion and the initial research question is suggested.
-
Methods:
-
Important information is missing, including details on the subjects' recruitment method, inclusion-exclusion criteria, ethical considerations, ethical approval, and the statistical techniques employed.
I kindly direct your attention to the attached PDF notes for a more detailed description of these suggestions.
Overall, with these adjustments, I believe the manuscript has the potential to make a significant contribution to the field. I appreciate your attention to these recommendations and look forward to seeing the revised version of the manuscript.

Author Response
Dear Reviewer,
We would like to thank you for your valuable comments on the article. Below you will find our reply to your review. All changes are with a description or a comment, and changes have been made to the manuscript (track changes in the tracking group on the review tab).
Comments and Suggestions for Authors
Dear editors,
Firstly, I would like to commend the authors on their intriguing and generally well-structured manuscript exploring the correlation between cigarette smoking and personality traits, anxiety, and the HINT1/rs2526303 polymorphism.
However, for this work to meet the standards required for publication, several improvements are essential. I have outlined the key areas that require attention below:
Introduction:
Addition of citations in specific sections is necessary to strengthen the scholarly foundation of the manuscript.
Thank you for these suggestions. Citations and full name of THC were added, the aim of the study was also reframed.
Discussion and Conclusions:
An extensive rearrangement of the material is recommended. The authors' findings and their correlation with relevant literature should form the central focus of this section. Extraneous theoretical or irrelevant information should be eliminated.
Establishing a clear connection between the conclusion and the initial research question is suggested.
Thank you for these valuable suggestions. All recommended changes were made in the Discussion and Conclusions sections.
Methods:
Important information is missing, including details on the subjects' recruitment method, inclusion-exclusion criteria, ethical considerations, ethical approval, and the statistical techniques employed.
Thank you for these valuable suggestions. All requested changes were made.
I kindly direct your attention to the attached PDF notes for a more detailed description of these suggestions.
Thank you for this additional input. All suggested changes were made in the manuscript.
Overall, with these adjustments, I believe the manuscript has the potential to make a significant contribution to the field. I appreciate your attention to these recommendations and look forward to seeing the revised version of the manuscript.
Round 2
Reviewer 1 Report
Comments and Suggestions for Authors
The authors revised the paper in accordance with my comments.
Author Response
Comments and Suggestions for Authors
The authors revised the paper in accordance with my comments.
Dear Reviewer,
We thank you for your valuable comments on the article and your contribution to improving it significantly.
Reviewer 2 Report
Comments and Suggestions for Authors
I would like to commend the authors for the significant improvements made since the initial submission. The dedication to enhancing the quality of the manuscript is evident, and the changes implemented have strengthened its overall presentation.
Specifically, I appreciate the attention given to the list of references, which has been enriched to enhance the scholarly context of the study. The results section is now clearly organized and presented, facilitating a more comprehensive understanding of the research findings. Furthermore, the re-arrangement of the discussion has significantly improved the clarity in illustrating the correlation between the research question, the obtained results, and the relevant literature cited.
While the manuscript has undergone notable improvements, I would like to emphasize the importance of including information regarding the method used for the recruitment of subjects in the study. A comprehensive understanding of the recruitment process is crucial for readers to assess the validity and generalizability of the study findings. Therefore, I kindly request that the authors provide detailed information on the recruitment methodology employed.
Comments on the Quality of English LanguageI noted minor spelling or grammar issues throughout the manuscript. I believe that a final proofreading would be sufficient to address these issues and ensure the overall linguistic quality of the manuscript.
Author Response
Dear Reviewer,
We would like to thank you for your valuable comments on the article. Below you will find our reply to your review. All changes are with a description or a comment, and changes have been made to the manuscript (track changes in the tracking group on the review tab).
Comments and Suggestions for Authors
I would like to commend the authors for the significant improvements made since the initial submission. The dedication to enhancing the quality of the manuscript is evident, and the changes implemented have strengthened its overall presentation.
Specifically, I appreciate the attention given to the list of references, which has been enriched to enhance the scholarly context of the study. The results section is now clearly organized and presented, facilitating a more comprehensive understanding of the research findings. Furthermore, the re-arrangement of the discussion has significantly improved the clarity in illustrating the correlation between the research question, the obtained results, and the relevant literature cited.
While the manuscript has undergone notable improvements, I would like to emphasize the importance of including information regarding the method used for the recruitment of subjects in the study. A comprehensive understanding of the recruitment process is crucial for readers to assess the validity and generalizability of the study findings. Therefore, I kindly request that the authors provide detailed information on the recruitment methodology employed.
Thank you for these valuable suggestions. Recommended changes were made in the Materials & Methods section.
Comments on the Quality of English Language
I noted minor spelling or grammar issues throughout the manuscript. I believe that a final proofreading would be sufficient to address these issues and ensure the overall linguistic quality of the manuscript.
Thank you for noticing the linguistic errors. The manuscript has been corrected.
Additionally, we would like to thank you for your valuable comments on the article and your unparalleled commitment and contribution to improving it significantly.